# Plasma taurine level is linked to symptom burden and clinical outcomes in post-COVID condition

Mobin Khoramjoo[1,2], Kaiming Wang[2,3¤], Karthik Srinivasan[4], Mahmoud Gheblawi[1,2], Rupasri Mandal[5], Simon Rousseau[6], David Wishart[5], Vinay Prasad[4], Lawrence Richer[7], Angela M. Cheung[8], Gavin Y. Oudit ![ORCID][2,3]*

1 Department of Physiology, University of Alberta, Edmonton, Alberta, Canada, 2 Mazankowski Alberta Heart Institute, University of Alberta, Edmonton, Alberta, Canada, 3 Division of Cardiology, Department of Medicine, University of Alberta, Edmonton, Alberta, Canada, 4 Department of Chemical and Materials Engineering, University of Alberta, Edmonton, Alberta, Canada, 5 The Metabolomics Innovation Center, University of Alberta, Edmonton, Alberta, Canada, 6 Department of Medicine, McGill University & The Research Institute of the McGill University Health Centre, Montreal, Canada, 7 Department of Pediatrics, Faculty of Medicine & Dentistry, University of Alberta, Edmonton, Alberta, Canada, 8 Department of Medicine, University Health Network, Toronto, Ontario, Canada

¤ Current address: Cumming School of Medicine, University of Calgary, Calgary, Canada
* gavin.oudit@ualberta.ca

**Data Availability Statement:** All relevant data are within the manuscript and its Supporting Information files.

## Abstract

### Background

A subset of individuals (10–20%) experience post-COVID condition (PCC) subsequent to initial SARS-CoV-2 infection, which lacks effective treatment. PCC carries a substantial global burden associated with negative economic and health impacts. This study aims to evaluate the association between plasma taurine levels with self-reported symptoms and adverse clinical outcomes in patients with PCC.

### Methods and findings

We analyzed the plasma proteome and metabolome of 117 individuals during their acute COVID-19 hospitalization and at the convalescence phase six-month post infection. Findings were compared with 28 age and sex-matched healthy controls. Plasma taurine levels were negatively associated with PCC symptoms and correlated with markers of inflammation, tryptophan metabolism, and gut dysbiosis. Stratifying patients based on the trajectories of plasma taurine levels during six-month follow-up revealed a significant association with adverse clinical events. Increase in taurine levels during the transition to convalescence were associated with a reduction in adverse events independent of comorbidities and acute COVID-19 severity. In a multivariate analysis, increased plasma taurine level between acute and convalescence phase was associated with marked protection from adverse clinical events with a hazard ratio of 0.13 (95% CI: 0.05–0.35; p<0.001).

**Funding:** For this study, GYO is supported by grants from the Canadian Institutes of Health Research (grant no. PJT-451105), the Northern Alberta Clinical Trials and Research Centre (grant no. RES50821), and the long COVID web (grant no. RES0065210) at the University of Alberta.

**Competing interests:** The authors have declared that no competing interests exist.

## Conclusions

Taurine emerges as a promising predictive biomarker and potential therapeutic target in PCC. Taurine supplementation has already demonstrated clinical benefits in various diseases and warrants exploration in large-scale clinical trials for alleviating PCC.

## Introduction

At least 10–20% of patients with prior SARS-CoV-2 infection continue to experience a variety of persistent or episodic symptoms, including fatigue, sleep disturbance, confusion, and dyspnea beyond three months from their initial infection, collectively known as the post-COVID condition (PCC) [1, 2]. Several pathophysiological mechanisms have been proposed to explain the persistence of symptoms including the presence of viral reservoir, sustained inflammation, endothelial dysfunction, accumulation of senescence cells, and impaired energy metabolism [1, 3, 4]. Epidemiological research has revealed an elevated risk of developing new diagnoses of pulmonary, cardiovascular, gastrointestinal, metabolic, psychiatric, and nervous system disorders associated with worse prognoses beyond one year post-infection [5–7]. Despite numerous investigations, an effective treatment for PCC remains elusive.

Metabolomics has provided valuable insights into the biomarkers and pathogenesis of COVID-19 and PCC, as SARS-CoV-2 infection dysregulates various metabolic pathways, including citric acid cycle, amino acid, lipid, and taurine and hypotaurine metabolism [1, 8–12]. Taurine (2–aminoethanesulfonic acid) is a semi-essential amino acid produced from cysteine through the enzymatic action of cysteine dioxygenase and cysteine sulfinic acid decarboxylase (CSAD) in mammalian cells [13]. Additionally, taurine is directly obtained from diet and is absorbed by cells through the taurine transporter to mediate critical biological functions by slowing cellular senescence, suppressing inflammation, and acting as an antioxidant [14–16].

In this study, we assessed the plasma metabolomic and proteomic profile of 117 individuals during the acute SARS-CoV-2 infection and six-month into the convalescence phase. Using state-of-the-art quantitative multi-omics, we identified a critical link between plasma taurine levels with PCC symptoms and adverse clinical outcomes, suggesting a potential protective role of taurine in alleviating the burdens of PCC.

## Methods

### Study participants

Participants included in this study were recruited through the COVID-19 Surveillance Collaboration (CoCollab) Study, which has been described previously [1, 10, 17]. Patients newly admitted to hospital wards designated for COVID-19 and intensive care units at the University of Alberta Hospital (Edmonton, Canada) between October 15, 2020, and June 29, 2021 were enrolled prospectively (**Fig 1A**). All enrolled patients were $\geq$ 18 years of age with a laboratory-confirmed COVID-19 diagnosis based on a positive SARS-CoV-2 real-time PCR (PCR) assay from nasopharyngeal swabs or lower respiratory samples. Comparisons were made against age and sex-matched healthy controls (n = 28) enrolled during the same period (**Table 1**).

### Plasma collection and storage

Venous blood sampling was performed in the morning by trained phlebotomists and transported to the Canadian Biosample Repository (CBSR) located at the University of Alberta

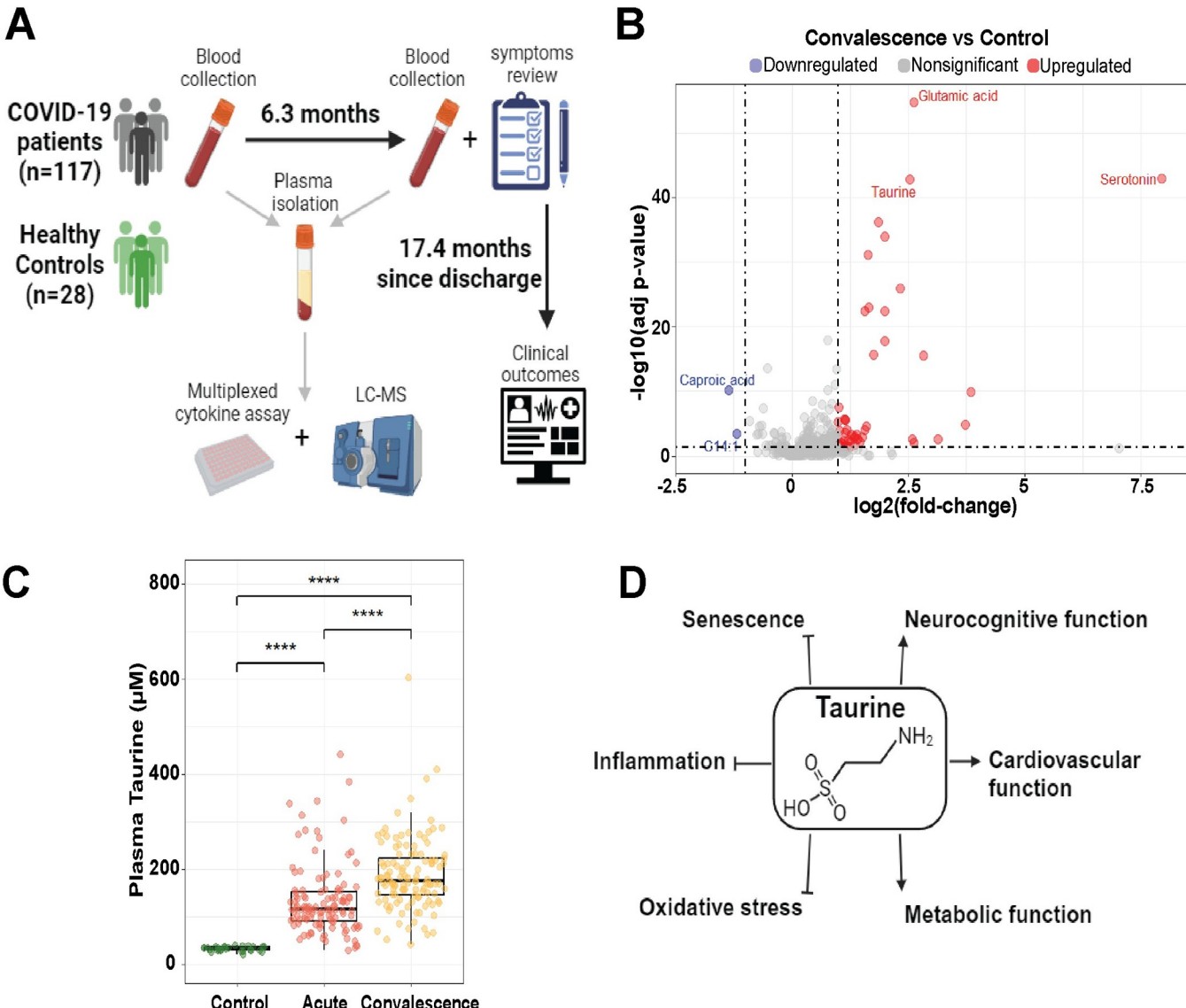

**Fig 1. Taurine in acute and convalescence phase of SARS-CoV-2 infection.** (A) Schematic of study design. (B) Differentially expressed metabolites between convalescence samples with age and sex-matched healthy controls. (C) Box and whisker plot showing taurine levels between cohorts. Mann-Whitney U test with Benjamini-Hochberg correction was performed for pairwise comparison. (D) Schematic representation of taurine mediated effects on various pathways and physiological processes. *p<0.05, **p<0.01, ***p<0.001, ****p<0.0001.

within one hour for immediate processing. Samples were collected in tubes containing ethyl-enediaminetetraacetic acid (EDTA) and centrifuged at 1500 x g for 10 min at room temperature. Plasma was subsequently aliquoted for storage at -80˚C. Baseline sampling of acute COVID-19 was performed immediately following hospital admission, while pre-scheduled follow-up blood sampling at six months was collected from ambulatory patients.

## Clinical outcomes and PCC symptom assessment

Detailed clinical characteristics, including demographics, vital signs, presenting symptoms, comorbidities, and medications were collected through individual review of electronic medical

**Table 1. Baseline clinical characteristics.**

| Variable | Overall n = 117[1] | Healthy controls n = 28[1] | Event free n = 81[1] | With event n = 36[1] | p-value[2] |
|---|---|---|---|---|---|
| Age | 62 (53, 73) | 55 (52, 59) | 62 (53, 73) | 63 (57, 75) | 0.6 |
| Male | 66 (56.4%) | 16 (57%) | 46 (57%) | 20 (56%) | >0.9 |
| BMI | 28 (24, 34) | - | 28 (25, 34) | 28 (24, 38) | >0.9 |
| Ethnicity | | - | | | 0.092 |
| Asian | 11 (9.4%) | - | 8 (9.9%) | 3 (8.3%) | |
| Caucasian | 75 (64%) | - | 54 (67%) | 21 (58%) | |
| East european | 6 (5.1%) | - | 3 (3.7%) | 3 (8.3%) | |
| Indian | 8 (6.8%) | - | 4 (4.9%) | 4 (11%) | |
| Indigenous | 3 (2.6%) | - | 0 (0%) | 3 (8.3%) | |
| Latin | 6 (5.1%) | - | 5 (6.2%) | 1 (2.8%) | |
| Middle east | 8 (6.8%) | - | 7 (8.6%) | 1 (2.8%) | |
| Severity | | | | | 0.042 |
| Recovered (no PCC symptoms) | 30 (26%) | - | 25 (31%) | 5 (14%) | |
| Mild (= <3 symptoms) | 32 (27%) | - | 24 (30%) | 8 (22%) | |
| Severe (>3 symptoms) | 55 (47%) | - | 32 (40%) | 23 (64%) | |
| WHO Scale | | | | | 0.7 |
| 3 | 16 (14%) | - | 10 (12%) | 6 (17%) | |
| 4 | 56 (48%) | - | 41 (51%) | 15 (42%) | |
| 5 | 26 (22%) | - | 19 (23%) | 7 (19%) | |
| 6 | 12 (10%) | - | 7 (8.6%) | 5 (14%) | |
| 7 | 7 (6.0%) | - | 4 (4.9%) | 3 (8.3%) | |
| Vaccination | 90 (77%) | - | 62 (77%) | 28 (78%) | >0.9 |
| SARS-CoV-2 Variant | | | | | >0.9 |
| Original | 85 (73%) | - | 57 (70%) | 28 (78%) | |
| B.1.1.7 | 25 (21%) | - | 18 (22%) | 7 (19%) | |
| B.1.351 | 1 (0.9%) | - | 1 (1.2%) | 0 (0%) | |
| B.1.617.2 | 1 (0.9%) | - | 1 (1.2%) | 0 (0%) | |
| P.1 | 5 (4.3%) | - | 4 (4.9%) | 1 (2.8%) | |

[1] Median (IQR); n (%)

[2] Kruskal-Wallis rank sum test; Pearson's Chi-squared test; Fisher's exact test. Abbreviations: BMI, body mass index; WHO, World Health Organization.

records. Incidence of all-cause mortality and hospital readmission since their discharge date from the acute COVID-19 hospitalization (median follow-up of 17.4 [IQR: 14.3–18.8] months) was obtained from individual's electronic medical records until June 30[th], 2022. A review of symptoms was performed using a questionnaire during follow-up blood sampling that contained general systemic, cardiopulmonary, neurological, and gastrointestinal domains.

## Multiplexed cytokine assays

Luminex xMAP technology was used to quantify 47 human cytokines, chemokines, and growth factors. The multiplexing analysis was performed using the Luminex 200 system (Luminex, Austin, TX, USA) by Eve Technologies Corp. (Calgary, Alberta). Forty-seven markers were concurrently measured from samples using Eve Technologies' Human Cytokine 48-Plex Discovery Assay (MilliporeSigma, Burlington, Massachusetts, USA). The assay was run according to the manufacturer's protocol. Assay sensitivities of these markers range from 0.14 to 55.8 pg/mL for the 48-plex.

## Targeted plasma proteomics by LC-MS

Targeted plasma proteomics was performed using liquid chromatography-mass spectrometry (LC-MS). This assay quantified 274 human proteins through a multiple reaction monitoring (MRM) approach with stable isotope-labelled standards (SIS). Each protein was represented by a carefully chosen peptide, employing the gold standard technique for LC-MRM-MS. Three levels of Quality control (QC) samples were monitored, ensuring accuracy throughout the process. Peptides were synthesized using FMOC chemistry techniques, and internal standards were used to compensate for variations. Sample preparation involved denaturation, reduction, alkylation, and proteolysis. LC-MRM-MS analysis was conducted after solid-phase extraction and rehydration. Standard curves were generated using natural isotopic abundance peptides, and QC samples were prepared accordingly. LC separation and mass spectrometry parameters were optimized for optimal peptide ionization and fragmentation. Quantitative analysis involved monitoring targets over cycles and detection windows, with data analysis performed using specialized software. Linear regression was used to calculate peptide concentrations in plasma samples based on the standard curve.

## Targeted plasma metabolomics by LC-MS

Targeted quantitative metabolomics using LC-MS was employed to analyze samples, combining direct injection mass spectrometry (DI-MS) and LC-MS/MS. This custom assay identified and quantified 635 endogenous metabolites, including amino acids, biogenic amines, lipids, organic acids, and nucleotides. Chemical derivatization, analyte extraction, and LC separation were used, with multiple reaction monitoring (MRM) pairs for detection. Isotope-labeled internal standards (ISTDs) and chemical derivatization standards ensured accurate quantification. Standard solutions were prepared by dissolving solids in water, and a 96-deep-well plate was used for sample preparation. Plasma samples underwent extraction, derivatization, and centrifugation before injection into the LC-MS system. For organic acid analysis, derivatization, and LC-MS injection were performed. Mass spectrometric analysis was conducted using an ABSciex 5500Qtrap tandem mass spectrometer. Data analysis utilized Analyst 1.6.2 software.

## Statistical analysis

Features (i.e., molecules) with >50% missing values were removed from all omic datasets. The remaining values below the lower limit of detection (LLOD) were imputed using the minimum value of each feature as previously described [1, 18, 19]. The dataset was normalized by applying a logarithmic (base10) transformation. Differential expression analysis with significant filtering criteria (p value<0.05 and fold-change>2) was performed using the corrected metabolites with MetaboAnalyst 5.0 (www.metaboanalyst.ca) [20] and visualized using R 4.2.3 (Vienna, Austria). Mann-Whitney U and paired Wilcoxon signed-rank tests were utilized for non-normalized distributions, while ANOVA was performed for parametric comparisons. p values were adjusted for multiple testing using the Benjamini and Hochberg false discovery rate (FDR) correction. Binary logistic regression was performed for the association between self-reported PCC symptoms and taurine levels in the convalescence phase, adjusting for age, sex, diabetes, chronic kidney disease (CKD), treatment received during acute COVID-19 (dexamethasone, antibiotics, tocilizumab, remdesivir), WHO Ordinal Scale for acute COVID-19 severity, and SARS-CoV-2 vaccination status. The Pearson correlation coefficient was employed to examine correlations between molecules and taurine levels in the convalescence phase. The association between taurine levels and adverse clinical outcomes was assessed using Cox proportional hazards regression and Kaplan-Meier survival analyses. Statistical analyses

were performed using R 4.2.3 (Vienna, Austria), with statistical significance considered based on 2-tailed p<0.05.

## Study approval

This study was conducted under the ethical principles of the Declaration of Helsinki with approval from the University of Alberta Health Research Ethics Board (Pro00100319_REN3). Written and informed consent was obtained from all participants.

## Results

### Patient characteristics

The median age of the overall cohort was 62 years (IQR: 53–73 years), with a near equivalent sex distribution (56.4% male), a marked prevalence of diabetes (42%), and predominance of infection by the ancestral SARS-CoV-2 strain (73%) (**Tables 1 and 2**). Patients were classified as event-free (n = 81) or with-event (n = 36) based on experience of adverse clinical outcomes following discharge from acute COVID-19 hospitalization. Patients with chronic kidney disease (CKD) were overrepresented in the with-event group (28%, p = 0.01) (**Table 2**).

### Temporal changes in taurine levels following SARS-CoV-2 infection

Metabolomic analysis of plasma samples in the convalescence phase identified 50 differentially expressed metabolites compared to age and sex-matched healthy controls, with the amino acid taurine as one of the top upregulated metabolites (fold-change = 5.7, p<0.0001) (**Fig 1B**). Additionally, plasma taurine levels were significantly elevated in acute infection compared to healthy controls (p = 1.1e-15) and were further elevated in the convalescence phase compared to the acute phase (p = 1.8e-11) (**Fig 1C**). Taurine could protect against PCC through inhibition of various pathophysiological processes and stimulation of beneficial functions (**Fig 1D**). Therefore, we hypothesize that failure to upregulate the taurine pathway may predispose individuals to having adverse outcomes and greater symptom burden during convalescence from SARS-CoV-2 infection.

**Table 2. Comorbidities and medications.**

| Variable | Overall, n = 117[1] | Controls n = 28[1] | Event free, n = 81[1] | With event, n = 36[1] | p-value[2] |
|---|---|---|---|---|---|
| HTN | 57 (49%) | - | 36 (44%) | 21 (58%) | 0.2 |
| DM2 | 49 (42%) | - | 30 (37%) | 19 (53%) | 0.11 |
| DLD | 36 (31%) | - | 25 (31%) | 11 (31%) | >0.9 |
| CKD | 18 (15%) | - | 8 (9.9%) | 10 (28%) | 0.013 |
| COPD/Asthma/OSA | 34 (29%) | - | 21 (26%) | 13 (36%) | 0.3 |
| CVA/CAD/PE | 24 (21%) | - | 13 (16%) | 11 (31%) | 0.073 |
| Supplemental O2 | 97 (83%) | - | 68 (84%) | 29 (81%) | 0.7 |
| Dexamethasone | 102 (87%) | - | 72 (89%) | 30 (83%) | 0.5 |
| Antibiotic | 88 (75%) | - | 61 (75%) | 27 (75%) | >0.9 |
| Tocilizumab | 17 (15%) | - | 13 (16%) | 4 (11%) | 0.5 |
| Remdesivir | 6 (5.1%) | - | 4 (4.9%) | 2 (5.6%) | >0.9 |

[1] n (%)

[2] Kruskal-Wallis rank sum test; Pearson's Chi-squared test; Fisher's exact test. Abbreviations: HTN, hypertension; DM2, diabetes mellitus type 2; DLD, dihydrolipoamide dehydrogenase; CKD, chronic kidney disease; COPD, Chronic obstructive pulmonary disease; OSA, Obstructive sleep apnea; CVA, cerebral vascular accident; CAD, coronary arterial disease; PE, pulmonary embolism.

## Correlation between taurine levels with inflammatory and microbiome biomarkers in post-COVID condition

We examined the associations between plasma taurine levels and biomarkers of pathophysiological processes identified in PCC during the convalescence phase. We found that plasma levels of taurine negatively correlated with markers of inflammation, such as C-reactive protein (R = -0.22, p = 0.019), interleukin-10 (R = -0.23, p = 0.014), and interleukin-6 (R = -0.28, p = 0.002) (**Fig 2A–2C**). In contrast, there was a positive correlation between taurine levels with tryptophan (R = 0.29, p = 0.001) and serotonin (R = 0.4, p = 8.8e-6), while demonstrating a strong negative correlation with quinolinic acid (r = -0.4, p<0.001), a byproduct of tryptophan metabolism with neurotoxic effects (**Fig 2D–2F**). Lastly, taurine was negatively correlated with biomarkers related to gut dysbiosis, such as trimethylamine N-oxide (TMAO) (R = -0.27, p = 0.003), phenylacetic acid (R = -0.22, p = 0.016), and lipopolysaccharide binding protein (R = -0.27, p = 0.0031), suggesting a possible role of taurine in modulating microbial ecology in PCC (**Fig 2G–2I**).

## Taurine levels are linked to self-reported symptoms in post-COVID condition

Taurine was one of the top five most altered metabolites between healthy controls, event-free, and with-event cohorts [False Discovery Rate (FDR) = 3.382E-46] (**Fig 3A**). Taurine levels were lower in patients with-event compared to those without event during convalescence

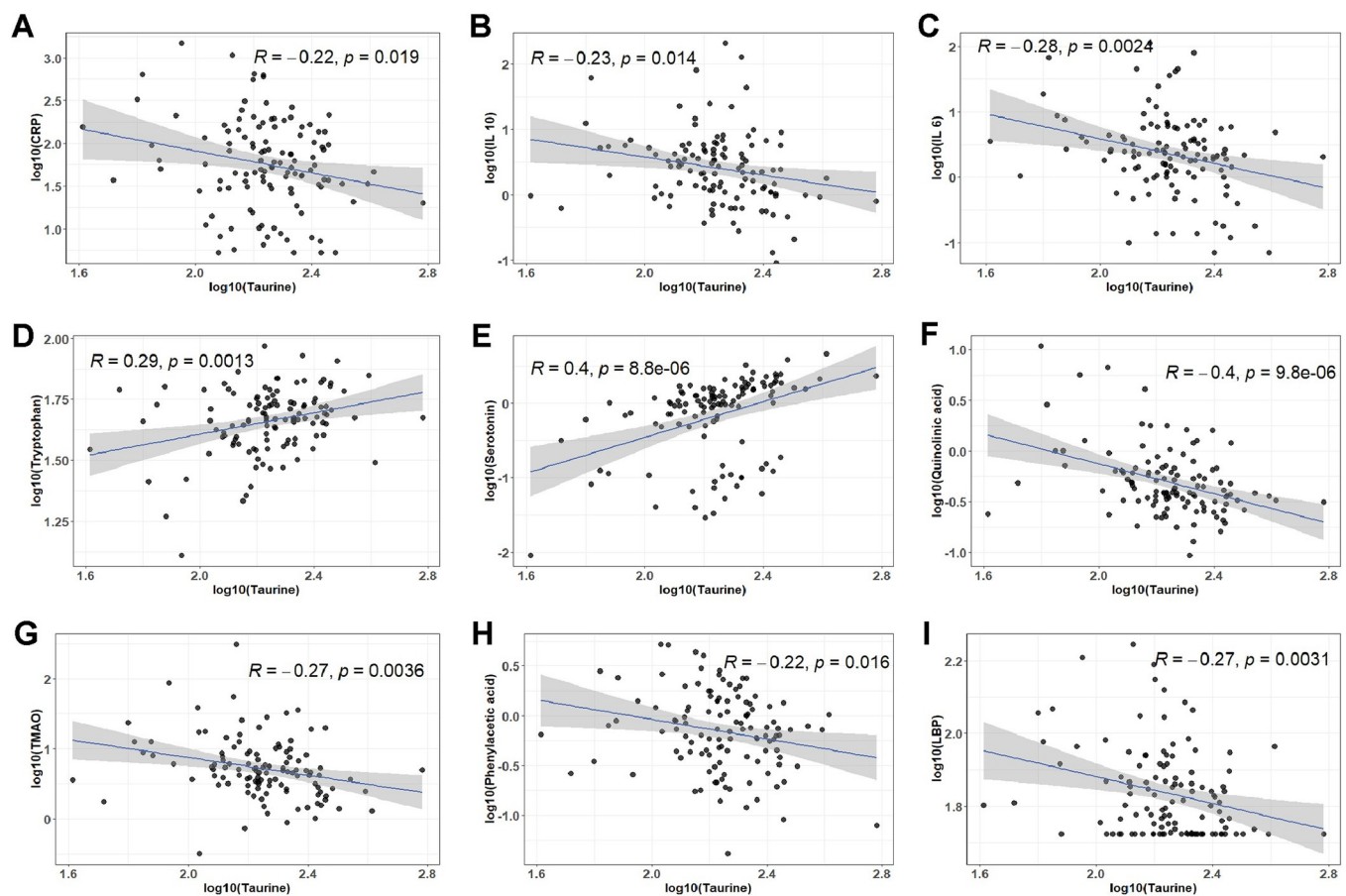

**Fig 2. Taurine correlates with biomarkers of inflammation and gut dysbiosis.** Scatter plots show the correlations between plasma taurine levels and plasma CRP (A), IL10 (B), IL6 (C), tryptophan (D), serotonin (E), quinolinic acid (F), TMAO (G), phenylacetic acid (H), and LBP (I) levels. CRP, C-reactive protein; IL10, Interleukin 10; IL6, Interleukin 6; TMAO, Trimethylamine N-oxide; LBP, Lipopolysaccharide binding protein.

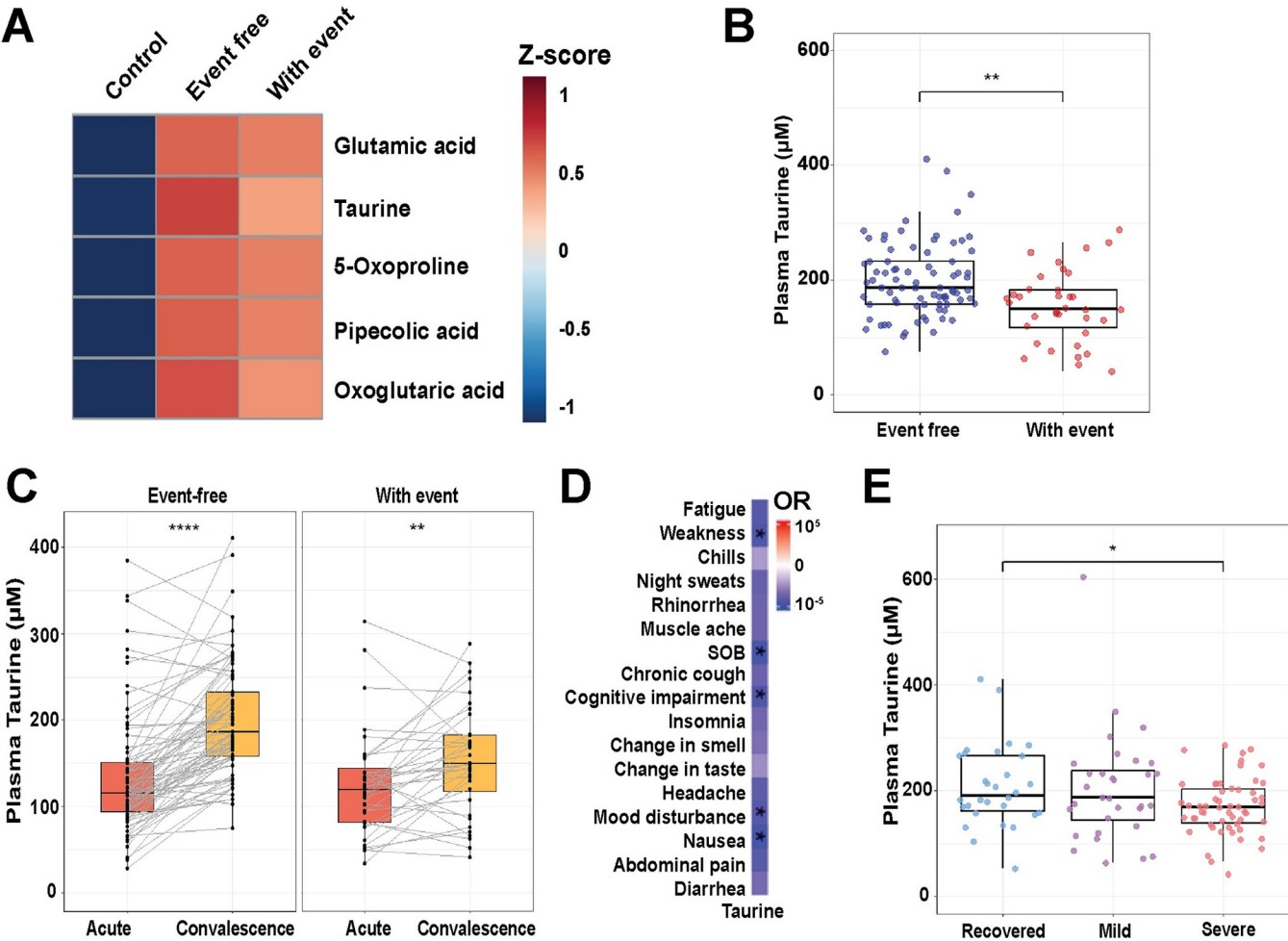

**Fig 3. Analysis of plasma taurine levels in the convalescence phase following SARS-CoV-2 infection.** (A) Heatmap showing the top five molecules with the most significant p values between patients with and without adverse events based on analysis of variance (ANOVA). (B) Box and whisker plot showing the level of taurine in event-free and with-event individuals. Mann-Whitney U test with Benjamini-Hochberg correction performed for pairwise comparison. (C) Paired box and whisker plots illustrating the trajectory of change in plasma taurine level from acute to convalescence phase in individual patients. Paired Wilcoxon signed-rank test was performed for pairwise comparison. (D) Heatmap showing the association between taurine levels with self-reported symptoms in the convalescence phase (OR: odds ratio) adjusted for clinical variables such as age, sex, diabetes, CKD, acute COVID-19 treatment (dexamethasone, antibiotics, tocilizumab, remdesivir), WHO Ordinal Scale for acute COVID-19 severity, and SARS-CoV-2 vaccination status. (E) Box and whisker plot illustrating taurine levels amongst PCC severity groups (severe: >3 symptoms, mild: ≦3 symptoms, recovered: no PCC symptoms). Mann-Whitney U test with Benjamini-Hochberg correction performed for pairwise comparison. *p<0.05, **p<0.01, ***p<0.001, ****p<0.0001.

(p = 0.001, **Fig 3B**). Concomitantly, patients who experienced adverse clinical events during follow-up were less likely to have a substantial rise in taurine levels from the acute phase (fold-change = 1.259, p = 0.0048) compared to event-free patients (fold-change = 1.616, p = 1.4E-12) (**Fig 3C**). During the convalescence phase, taurine concentrations were negatively associated with PCC symptoms, including weakness, shortness of breath, cognitive impairment, mood disturbance, and nausea (**Fig 3D**) and were significantly lower in patients with greater symptom burden (severe: > 3 symptoms) compared to individuals reporting complete resolution of symptoms (recovered) (**Fig 3E**).

## Plasma taurine levels are linked to clinical outcomes during convalescence

Patients were hence stratified based on the trajectories of plasma taurine levels from the acute infection to convalescence. In 97 individuals, taurine levels notably increased (p<2e-16), while

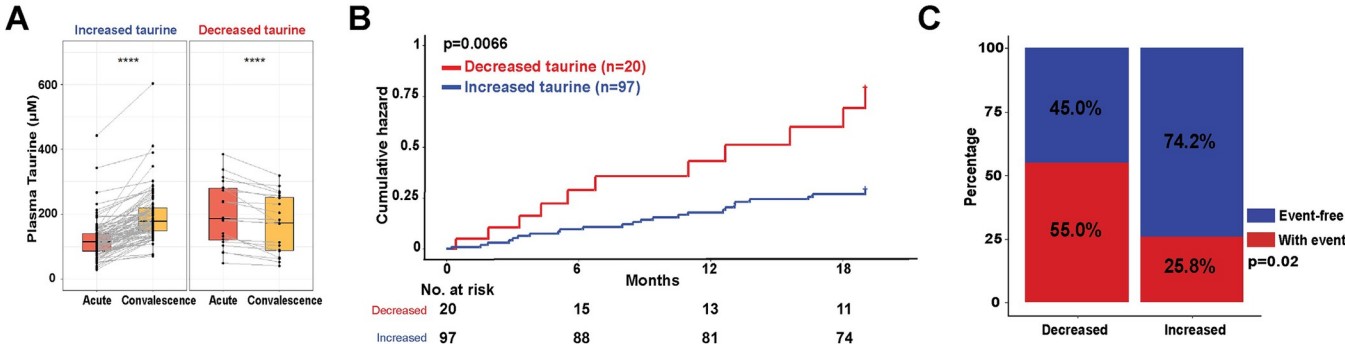

**Fig 4. Relationship between temporal changes in plasma taurine and adverse clinical outcomes.** (A) Paired box and whisker plots illustrating the trajectory of plasma taurine level from acute to convalescence phase. Paired Wilcoxon signed-rank test was performed for pairwise comparison. (B) Association between the trajectory of plasma taurine levels with adverse clinical outcomes (all-cause mortality and re-hospitalization) following discharge from acute COVID-19 hospitalization. Log-rank p = 0.0066. (C) Stack bar plot showing the proportion of event-free and with-event individuals between patients with either reduced or increased taurine levels from acute to convalescence phase. Chi-square test. P = 0.02. *p<0.05, **p<0.01, ***p<0.001, ****p<0.0001.

20 patients exhibited a significant reduction during this transition (p = 9.6e-5) (**Fig 4A**). The Kaplan-Meier survival analysis revealed a significant elevation in the cumulative hazard for patients with reduction in taurine levels compared to those with increase in taurine levels from the acute to convalescence phase (log-rank p = 0.0066) (**Fig 4B**). A greater proportion of patients whose taurine levels decreased during the convalescence period eventually encountered an adverse clinical event (55.0%) compared to those with increased taurine levels (25.8%) (p = 0.02) (**Fig 4C**). In the multivariate time-to-event analysis, increased taurine levels between the acute and convalescence phase was significantly associated with a reduction in event risk with an adjusted hazard ratio of 0.13 (p<0.001, 95% CI:0.05–0.35) (**Fig 5**). In comparison, presence of chronic kidney disease (CKD) (aHR:4.15, 95% CI:1.71–10.07) and greater acute COVID-19 severity according to the WHO scale (aHR:1.96, 95% CI:1.24–3.08) were associated with a higher risk of adverse clinical events during convalescence (**Fig 5**).

## Plasma taurine levels in patients with diabetes

Plasma taurine levels were examined in PCC patients with diabetes during the convalescence phase. Notably, PCC patients with diabetes exhibited reduced plasma taurine concentrations compared to PCC patients without diabetes (p<0.05) (**Fig 6A**). Further exploration of plasma samples from PCC patients with diabetes revealed associations between taurine and several key biomarkers involved in different biological functions. Plasma taurine levels were positively correlated with anti-inflammatory cytokines, namely interleukin-4 (R = 0.51, p<0.001) and interleukin-13 (R = 0.4, p = 0.0046) (**Fig 6B and 6C**). Similarly, plasma taurine levels positively correlated with antithrombin-3 (R = 0.38, p = 0.0064) and peroxiredoxin-1 (R = 0.62, p<0.001), which are involved in the inhibition of coagulation and oxidative stress, respectively (**Fig 6D and 6E**). Moreover, plasma taurine levels showed a positive correlation with apolipoprotein A1 (APO A-I) (R = 0.29, p = 0.043) in PCC patients with diabetes, which illustrates a potential role for taurine in lipid metabolism and cholesterol balance (**Fig 6F**). Moreover, we found that plasma taurine levels were positively associated with creatine (R = 0.32, p = 0.026), a source of energy in the body (**Fig 6G**). Lastly, taurine levels were negatively correlated with quinolinic acid (R = -0.3, p = 0.038), a neurotoxic breakdown metabolite of tryptophan (**Fig 6H**).

## Discussion

The COVID-19 pandemic has left a lasting impact resulting in a substantial burden for patients, the healthcare system, and our economy. Extensive research is underway to examine

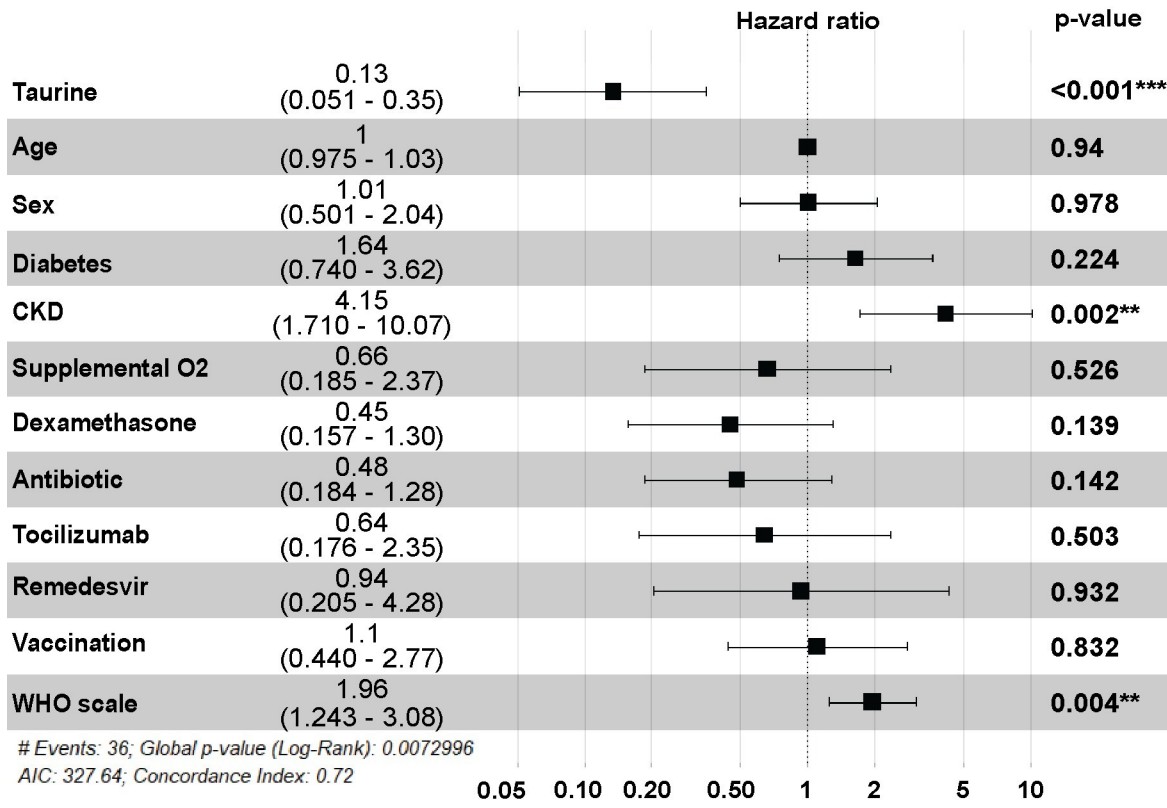

**Fig 5. Multivariate analysis of predictive factors for adverse clinical outcome.** Forest plot showing the adjusted hazard ratios, confidence intervals, and p values for plasma taurine level along with other variables such as age, sex, diabetes, CKD, acute COVID-19 treatment (dexamethasone, antibiotics, tocilizumab, remdesivir), WHO Ordinal Scale for acute COVID-19 severity, and SARS-CoV-2 vaccination status.

the underlying mechanisms and potential treatments of PCC. Our study focused on the link between plasma taurine levels in 117 patients during the acute and convalescence phase of SARS-CoV-2 infection with serum biomarkers measured using multi-omics platforms. Most individuals were recruited during Canada's second and third wave of the pandemic when the dominant SARS-CoV-2 strain was the original and B.1.1.7 variant [1, 10, 17]. In the repeat sampling at 6.3 months (IQR: 6.0–7.1) months, 74% of participants had at least one symptom related to PCC. Our study revealed a clear relationship between plasma taurine levels with metabolic alterations, PCC symptoms, and adverse clinical outcomes, highlighting a potential protective role of taurine against the PCC.

In various animal studies, taurine administration has shown benefits such as reducing depressive behavior, improving memory, and mitigating age-related issues by addressing cellular senescence, chronic inflammation, DNA damage, and mitochondrial dysfunction [14, 21, 22]. Similar beneficial effects have also been observed in clinical studies, where taurine supplementation significantly improved the cognitive function of patients with dementia, first-episode psychotic disorder, and mitochondrial encephalomyopathy with lactic acidosis and stroke-like episodes (MELAS) [23–25]. Consistently, our study showed higher plasma taurine level was correlated with various biomarkers reflecting improved physiological states, along with negative associations with different self-reported PCC symptoms, such as weakness, cognitive impairment, and mood disturbance.

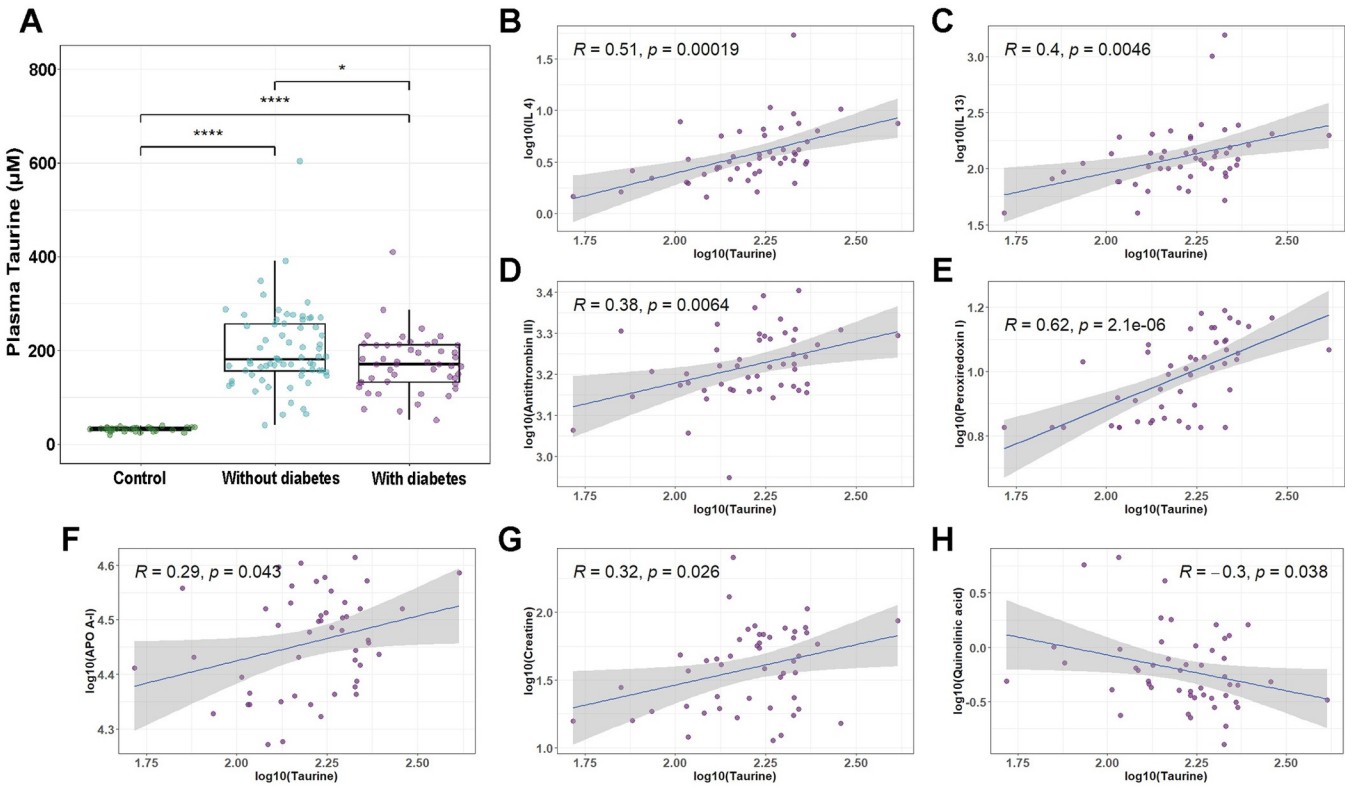

**Fig 6. Plasma taurine alteration and correlations with biomarkers in PCC patients with diabetes.** (A) Box and whisker plot illustrating the level of taurine in PCC patients with and without diabetes during the convalescence phase compared with healthy controls. Mann-Whitney U test with Benjamini-Hochberg correction was performed for pairwise comparison. (B-H) Scatter plots showing the correlations between plasma taurine levels with IL-4 (B), IL-13 (C), antithrombin III (D), peroxiredoxin I (E), APOA-I (F), creatine (G), and quinolinic acid (H) in PCC patients with diabetes. IL-4, Interleukin-4; IL-13, Interleukin-13; APOA-I, apolipoprotein A1. *p<0.05, **p<0.01, ***p<0.001, ****p<0.0001.

Taurine supplementation can decrease blood pressure, as well as total cholesterol and tri-glycerides in individuals with obesity, hypertension, and chronic hepatitis [26–28]. Many clinical trials have demonstrated the beneficial effects of taurine supplementation for patients with diabetes by improving glycemic indices and lipid profile [29–33]. Our study illustrated that PCC patients with comorbid diabetes had lower taurine levels compared to PCC patients without diabetes. However, taurine levels in these patients were positively correlated with molecules with anti-oxidation, anti-inflammatory, and cytoprotective properties. As such, taurine supplementation may be an effective adjuvant therapy for PCC patients with concurrent diabetes.

Recent studies have highlighted a link between taurine deficiency and accelerated cellular senescence, whereas supplementation with taurine reversed the negative effects [14, 34]. Senescence worsens the PCC and removing senescent cells reduced SARS-CoV-2 viral load alongside pulmonary and systemic inflammation [3]. These intriguing findings further support a potential role of taurine in alleviating PCC, given its biological functions to mitigate inflammation, oxidation, and cellular senescence, which are core pathophysiological processes underlying COVID-19 and the PCC. Gut dysbiosis has been identified as another pathophysiological mechanism in the PCC [2, 35]. Emerging evidence found alterations in gut microbiome of patients with PCC, resulting in the accumulation of gut-derived metabolites in the systemic circulation and resultant clinical manifestation of gastrointestinal and neurological symptoms [1, 36]. Taurine alleviates these conditions through different mechanisms, including regulation

of gut microbiota composition and enhancing tight junction proteins, thereby reducing gut permeability [37–40]. In accordance, we found that plasma taurine levels were negatively correlated with gut-derived metabolites, such as TMAO, quinolinic acid, and phenylacetic acid in PCC patients during convalescence. Another pathophysiological link between PCC and gut function is through tryptophan dysregulation. PCC is associated with decreased intestinal absorption of serotonin precursor, tryptophan, which impacts downstream serotonin metabolism and availability, possibly explaining certain neurocognitive symptoms in PCC patients [41]. Interestingly, taurine has a positive effect on serotonin levels, which has been demonstrated in both *in vitro* and *in vivo* studies [42, 43]. Consistent with these findings, plasma taurine levels in PCC patients were positively correlated with serotonin concentration and its precursor tryptophan during convalescence.

Our findings specifically reveal a robust association between hampered elevation of plasma taurine during the convalescence phase of SARS-CoV-2 infection with greater symptom burden and worse clinical outcomes. Other studies have also reported reduced taurine levels in patients with severe PCC symptoms up to 20 months after the initial infection [12, 44]. Similarly, in patients with myalgic encephalomyelitis/chronic fatigue syndrome (ME/CFS), taurine is reduced, highlighting the role of taurine in skeletal muscle, central nervous system, and energy homeostasis that could partly explain why so many patients experience fatigue in these two diseases [21, 45]. Indeed, taurine deficiency worsens skeletal muscle dysfunction and fatigue that is improved with taurine supplementation, which is consistent with recent findings in skeletal muscle biopsies from patients with PCC [4, 46, 47].

In conclusion, we demonstrated that taurine plays a critical role in the manifestation of symptoms and incidence of adverse clinical outcomes in the PCC. Taurine supplementation has demonstrated diverse therapeutic properties, including anti-oxidation, anti-aging, antiepileptic, cytoprotective, and cardioprotective effects in many diseases, including diabetes, obesity, and dilated cardiomyopathy (DCM) [21, 48–52]. Taurine is currently approved for heart failure treatment in Japan, and no toxicity has been reported in this context of clinical use [53]. Our findings highlight the need for large-scale clinical trials to determine the efficacy of taurine supplementation in alleviating symptom burden and improving outcomes for patients with PCC [26, 27].

## Supporting information

**S1 Checklist. Human participants research checklist.**
(DOCX)

**S1 File. Supporting data values of all plots.**
(XLSX)

## Acknowledgments

We would like to thank the patients, their families, and the dedicated clinical staff and frontline workers at the University of Alberta's COVID-19 units, whose collaboration made this work possible. Special appreciation goes to Dr. Bruce Ritchie and the Canadian Biosample Repository team for their noteworthy efforts in contributing to the CoCollab COVID-19 Study. The metabolomics assays were carried out by The Metabolomics Innovation Center in Edmonton, AB, Canada.

## Author Contributions

**Conceptualization:** Mobin Khoramjoo, Gavin Y. Oudit.

**Data curation:** Kaiming Wang, Mahmoud Gheblawi.

**Formal analysis:** Mobin Khoramjoo.

**Methodology:** Rupasri Mandal, David Wishart.

**Project administration:** Gavin Y. Oudit.

**Supervision:** Gavin Y. Oudit.

**Visualization:** Mobin Khoramjoo.

**Writing – original draft:** Mobin Khoramjoo.

**Writing – review & editing:** Kaiming Wang, Karthik Srinivasan, Mahmoud Gheblawi, Rupasri Mandal, Simon Rousseau, David Wishart, Vinay Prasad, Lawrence Richer, Angela M. Cheung, Gavin Y. Oudit.

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
