## [Decision Letter · Decision Letter 0]

3 Apr 2024

PONE-D-24-02915Linking Plasma Taurine Levels with Post-COVID Outcomes Reveals Therapeutic ImplicationsPLOS ONE

Dear Dr. Oudit,

Thank you for submitting your manuscript to PLOS ONE. After careful consideration, we feel that it has merit but does not fully meet PLOS ONE’s publication criteria as it currently stands. Therefore, we invite you to submit a revised version of the manuscript that addresses the points raised during the review process.

We look forward to receiving your revised manuscript.

Kind regards,

Anand Thirupathi

Academic Editor

PLOS ONE

Journal Requirements:

"For this study, GYO is supported by grants from the Canadian Institutes of Health Research (grant no. PJT-451105), the Northern Alberta Clinical Trials and Research Centre (grant no. RES50821), and the long COVID web (grant no. RES0065210) at the University of Alberta."

"This work is supported by grants from the Canadian Institutes of Health Research (grant no. PJT-451105), the Northern Alberta Clinical Trials and Research Centre (grant no. RES50821), and the long COVID web (grant no. RES0065210) at the University of Alberta."

"For this study, GYO is supported by grants from the Canadian Institutes of Health Research (grant no. PJT-451105), the Northern Alberta Clinical Trials and Research Centre (grant no. RES50821), and the long COVID web (grant no. RES0065210) at the University of Alberta."

**Additional Editor Comments:**

Dear Authors,

However both reviewers have suggested minor revisions, the comments raised by both reviewers during the review process is very important. Therefore, the authors must carry out the revisions carefully according to the reviewers comments before the paper is accepting.

Reviewers' comments:

Reviewer's Responses to Questions

**Comments to the Author**

1. Is the manuscript technically sound, and do the data support the conclusions?

Reviewer #1: Yes

Reviewer #2: Yes

2. Has the statistical analysis been performed appropriately and rigorously? 

Reviewer #1: Yes

Reviewer #2: Yes

3. Have the authors made all data underlying the findings in their manuscript fully available?

Reviewer #1: Yes

Reviewer #2: Yes

4. Is the manuscript presented in an intelligible fashion and written in standard English?

Reviewer #1: Yes

Reviewer #2: Yes

5. Review Comments to the Author

Reviewer #1: Authors should provide data which can support the taurine administration and its benefits like reduced depressive

behavior, improved memory, and mitigation of age-related issues. The authors are expected to study the correlation between the plasma taurine levels and therapeutic activities, including antioxidant, anti-aging, antiepileptic, cytoprotective, and cardioprotective effects in patients having diseases like diabetes, obesity. It will provide a scope for researches in this domain in future.

Authors can contribute some substantial findings to support the taurine supplementation to reduce the total cholesterol and triglycerides in patients with obesity.

Reviewer #2: This study aimed to investigate the role of taurine and possible recommendations of taurine supplementation for post-COVID conditions. However, the submitted article needs a thorough revision in order to bring better quality of scientific presentation towards the easy understanding of the article. The authors are advised to respond thoroughly to all the corrections mentioned below. I recommend this manuscript for publication in PLosOne, once the authors consider the revisions for the below comments.

This study aimed to investigate the role of taurine and possible recommendations of taurine supplementation for post-COVID conditions. Although the manuscript is well written, the results sections were quite confusing for readers. For example, the authors explained the data with repetition (“Fig 1. Taurine in acute and convalescence phase of SARS-CoV-2 infection.” Fig 2. Taurine correlations with biomarkers of inflammation and gut dysbiosis). This must be rewritten.

The authors explained about gut dysbiosis. However, it needs to be explained with the taurine supplementation with gut dysbiosis markers in the PCC. In addition, authors explaining about the role of taurine and tryptophan could strengthen these results.

6. PLOS authors have the option to publish the peer review history of their article (what does this mean?). If published, this will include your full peer review and any attached files.

Reviewer #1: **Yes: **Natarajan Sampathkumar

Reviewer #2: No

---

## [Author Response · Author response to Decision Letter 0]

22 Apr 2024

Response to the editor: 

Response: We would like to thank the editor for their effort towards our manuscript. We made sure that our manuscript meets the style requirements found in the links. 

"This work is supported by grants from the Canadian Institutes of Health Research (grant no. PJT-451105), the Northern Alberta Clinical Trials and Research Centre (grant no. RES50821), and the long COVID web (grant no. RES0065210) at the University of Alberta."

"For this study, GYO is supported by grants from the Canadian Institutes of Health Research (grant no. PJT-451105), the Northern Alberta Clinical Trials and Research Centre (grant no. RES50821), and the long COVID web (grant no. RES0065210) at the University of Alberta."

Response: We thank the editor for this comment. We removed the funding-related text from the acknowledgement section. This section reads as follows now: “We would like to thank the patients, their families, and the dedicated clinical staff and frontline workers at the University of Alberta's COVID-19 units, whose collaboration made this work possible. Special appreciation goes to Dr. Bruce Ritchie and the Canadian Biosample Repository team for their noteworthy efforts in contributing to the CoCollab COVID-19 Study. The metabolomics assays were carried out by The Metabolomics Innovation Center in Edmonton, AB, Canada.” 

Please update our funding statement section to this text: 

For this study, GYO is supported by grants from the Canadian Institutes of Health Research (grant no. PJT-451105), the Northern Alberta Clinical Trials and Research Centre (grant no. RES50821), and the Long COVID web (grant no. RES0065210) at the University of Alberta.

Response: We thank the editor for this comment. We have added the caption for supporting information file at the end of our manuscript. 

Changes made: 

Supporting information

S1 File. Supporting data values of all plots.

Response: We ensured that all references used in this manuscript are complete and correct. Vancouver style used for this section as requested in the author’s guideline. We also made sure that there is no retracted article cited.

Response: We have checked all the figures by Preflight Analysis and Conversion Engine (PACE) to ensure they all meet the PLOS requirements. 

Response to reviewers:

Reviewer #1: Authors should provide data which can support the taurine administration and its benefits like reduced depressive behavior, improved memory, and mitigation of age-related issues. 

Response: We would like to thank the reviewer for their time and effort towards our manuscript. As we have elaborated in the discussion section, taurine administration enhances various biological processes, resulting in beneficial neurological functions, such as reduced depression, improved memory, and alleviation of age-related issues. We have now cited several references that show taurine supplementation could enhance the cognitive function in different disorders. This is consistent with our findings where plasma taurine was positively correlated with different biomarkers involved in those biological processes and negatively associated with neurological symptoms in patients with post-COVID condition. Please note that in this study, we just examined the plasma taurine levels in patients with PCC during the acute and convalescence phase, and we did not supplement the patients with taurine. At the end of the discussion, we also mentioned that a large clinical trial is required to test these beneficial functions of taurine in PCC patients directly. 

Changes made: 

Page 16, line 323 (Discussion): “Similar beneficial effects have also been observed in clinical studies, where taurine supplementation significantly improved the cognitive function of patients with dementia, first-episode psychotic disorder, and Mitochondrial encephalomyopathy with lactic acidosis and stroke-like episodes (MELAS).(23-25) Consistently, our study showed higher plasma taurine level was correlated with various biomarkers reflecting improved physiological states, along with negative associations with different self-reported PCC symptoms, such as weakness, cognitive impairment, and mood disturbance.”

The authors are expected to study the correlation between the plasma taurine levels and therapeutic activities, including antioxidant, anti-aging, antiepileptic, cytoprotective, and cardioprotective effects in patients having diseases like diabetes, obesity. It will provide a scope for researches in this domain in future.

Response: We thank the reviewer for this comment. We agree that taurine can have more potent beneficial effects on PCC patients who are comorbid with other diseases, such as diabetes, obesity, or hypertension. To evaluate this, we assessed plasma taurine levels in patients with other comorbidities during the convalescence phase of COVID-19 disease. Out of all the comorbidities reported in Table 2, we could find that plasma taurine levels were reduced in PCC patients with diabetes compared to PCC patients without diabetes. We also demonstrated that taurine significantly correlates with other molecules with antioxidation, anti-inflammation, and cryoprotective properties in PCC patients with diabetes. These findings are now presented in the result and discussion sections. 

Changes made: 

Page 14, line 283 (Results, section 6): “Plasma taurine levels in patients with diabetes

Plasma taurine levels were examined in PCC patients with diabetes during the convalescence phase. Notably, PCC patients with diabetes exhibited reduced plasma taurine concentrations compared to PCC patients without diabetes (p<0.05) (Fig 6A). Further exploration of plasma samples from PCC patients with diabetes revealed associations between taurine and several key biomarkers involved in different biological functions. Plasma taurine levels were positively correlated with anti-inflammatory cytokines, namely interleukin-4 (R=0.51, p<0.001) and interleukin-13 (R=0.4, p=0.0046) (Fig 6B-C). Similarly, plasma taurine levels positively correlated with antithrombin-3 (R=0.38, p=0.0064) and peroxiredoxin-1 (R=0.62, p<0.001), which are involved in the inhibition of coagulation and oxidative stress, respectively (Fig 6D-E). Moreover, plasma taurine levels showed a positive correlation with apolipoprotein A1 (APO A-I) (R=0.29, p=0.043) in PCC patients with diabetes, which illustrates a potential role for taurine in lipid metabolism and cholesterol balance (Fig 6F). Moreover, we found that plasma taurine levels were positively associated with creatine (R=0.32, p=0.026), a source of energy in the body (Fig 6G). Lastly, taurine levels were negatively correlated with quinolinic acid (R=-0.3, p=0.038), a neurotoxic breakdown metabolite of tryptophan (Fig 6H).

Fig 6. Plasma taurine alteration and correlations with biomarkers in PCC patients with diabetes. 

(A) Box and whisker plot illustrating the level of taurine in PCC patients with and without diabetes during the convalescence phase compared with healthy controls. Mann-Whitney U test with Benjamini-Hochberg correction was performed for pairwise comparison. (B-H) Scatter plots showing the correlations between plasma taurine levels with IL-4 (B), IL-13 (C), antithrombin III (D), peroxiredoxin I (E), APOA-I (F), creatine (G), and quinolinic acid (H) in PCC patients with diabetes. IL-4, Interleukin-4; IL-13, Interleukin-13; APOA-I, apolipoprotein A1. *p<0.05, **p<0.01, ***p<0.001, ****p<0.0001.”

Page 16, line 332 (Discussion): “Many clinical trials have demonstrated the beneficial effects of taurine supplementation for patients with diabetes by improving glycemic indices and lipid profile. (29-33) Our study illustrated that PCC patients with comorbid diabetes had lower taurine levels compared to PCC patients without diabetes. However, taurine levels in these patients were positively correlated with molecules with anti-oxidation, anti-inflammatory, and cytoprotective properties. As such, taurine supplementation may be an effective adjuvant therapy for PCC patients with concurrent diabetes.”

Authors can contribute some substantial findings to support the taurine supplementation to reduce the total cholesterol and triglycerides in patients with obesity.

Response: As discussed in the previous comment, we cannot directly show that taurine supplementation will reduce total cholesterol and triglycerides in patients with obesity since we did not supplement our patients with taurine. We were able to examine the relationship between changes in cholesterol and triglyceride levels with taurine during follow-up. However, as evident by the figures below, we did not observe a significant difference in plasma taurine levels with and without obesity, which can result from the limited study cohort size. This point was explained in the discussion section to highlight the beneficial effects of taurine in other diseases through various biological functions, which can also help patients with post-COVID condition.

Figure 1. Plasma Taurine levels in PCC patients during convalescence.

(A). Plasma taurine levels in healthy controls, PCC patients with obesity (BMI>30), and PCC patients without obesity (BMI<30).

(B). Plasma taurine levels in healthy controls, diabetic obese PCC patients, and PCC patients without obesity and diabetes.

Reviewer #2: 

This study aimed to investigate the role of taurine and possible recommendations of taurine supplementation for post-COVID conditions. Although the manuscript is well written, the results sections were quite confusing for readers. For example, the authors explained the data with repetition (“Fig 1. Taurine in acute and convalescence phase of SARS-CoV-2 infection.” Fig 2. Taurine correlations with biomarkers of inflammation and gut dysbiosis). This must be rewritten.

Response: We thank the reviewer for their valuable input and comments. In the revised manuscript, we have now separated these two figures into two different sections in the result to offer more clarity for readers. We also rewrote some sentences in this part of the manuscript so that it will read better.

Changes made: 

Page 10, line 184 (Results, section 2): “Temporal changes in taurine levels following SARS-CoV-2 infection 

Metabolomic analysis of plasma samples in the convalescence phase identified 50 differentially expressed metabolites compared to age and sex-matched healthy controls, with the amino acid taurine as one of the top upregulated metabolites (fold-change=5.7, p<0.0001) (Fig 1B). Additionally, plasma taurine levels were significantly elevated in acute infection compared to healthy controls (p=1.1e-15) and were further elevated in the convalescence phase compared to the acute phase (p=1.8e-11) (Fig 1C). Taurine could protect against PCC through inhibition of various pathophysiological processes and stimulation of beneficial functions (Fig 1D). Therefore, we hypothesize that failure to upregulate the taurine pathway may predispose individuals to having adverse outcomes and greater symptom burden during convalescence from SARS-CoV-2 infection.”

Page 11, line 202 (Results, section 3): “Correlation between taurine levels with inflammatory and microbiome biomarkers in post-COVID condition

We examined the associations between plasma taurine levels and biomarkers of pathophysiological processes identified in PCC during the convalescence phase. We found that plasma levels of taurine negatively correlated with markers of inflammation, such as C-reactive protein (R=-0.22, p=0.019), interleukin-10 (R=-0.23, p=0.014), and interleukin-6 (R=-0.28, p=0.002) (Fig 2A-C). In contrast, there was a positive correlation between taurine levels with tryptophan (R=0.29, p=0.001) and serotonin (R=0.4, p=8.8e-6), while demonstrating a strong negative correlation with quinolinic acid (r=-0.4, p<0.001), a byproduct of tryptophan metabolism with neurotoxic effects (Fig 2D-F). Lastly, taurine was negatively correlated with biomarkers related to gut dysbiosis, such as trimethylamine N-oxide (TMAO) (R=-0.27, p=0.003), phenylacetic acid (R=-0.22, p=0.016), and lipopolysaccharide binding protein (R=-0.27, p=0.0031), suggesting a possible role of taurine in modulating microbial ecology in PCC (Fig 2G-I).”

Correlation between taurine levels with inflammatory and microbiome biomarkers in post-COVID condition

We examined the associations between plasma taurine levels and biomarkers of pathophysiological processes identified in PCC during the convalescence phase. We found that plasma levels of taurine negatively correlated with markers of inflammation, such as C-reactive protein (R=-0.22, p=0.019), interleukin-10 (R=-0.23, p=0.014), and interleukin-6 (R=-0.28, p=0.002) (Fig 2A-C). In contrast, there was a positive correlation between taurine levels with tryptophan (R=0.29, p=0.001) and serotonin (R=0.4, p=8.8e-6), while demonstrating a strong negative correlation with quinolinic acid (r=-0.4, p<0.001), a byproduct of tryptophan metabolism with neurotoxic effects (Fig 2D-F). Lastly, taurine was negatively correlated with biomarkers related to gut dysbiosis, such as trimethylamine N-oxide (TMAO) (R=-0.27, p=0.003), phenylacetic acid (R=-0.22, p=0.016), and lipopolysaccharide binding protein (R=-0.27, p=0.0031), suggesting a possible role of taurine in modulating microbial ecology in PCC (Fig 2G-I).

Page 10, line 190 (Results, section 2): 

---

## [Editor Report · Decision Letter 1]

14 May 2024

Plasma taurine level is linked to symptom burden and clinical outcomes in post-COVID condition

PONE-D-24-02915R1

Dear Dr. Oudit

We’re pleased to inform you that your manuscript has been judged scientifically suitable for publication and will be formally accepted for publication once it meets all outstanding technical requirements.

Kind regards,

Anand Thirupathi

Academic Editor

PLOS ONE
---

## [Editor Report · Acceptance letter]

27 May 2024

PONE-D-24-02915R1 

PLOS ONE

Dear Dr. Oudit, 

I'm pleased to inform you that your manuscript has been deemed suitable for publication in PLOS ONE. Congratulations! Your manuscript is now being handed over to our production team.

Kind regards, 

on behalf of

Dr. Anand Thirupathi 

Academic Editor

PLOS ONE